The function and mechanisms of action of lysyl oxidase-like 3 (LOXL3) in cancers

Zhao Dan 1
Su Pu 2
Peng Xuan 3
Cheng Xue 4
Li Bin 1
Tang Xi-min 1
Huang Shaoyang 5
Li Zhengliang 4
Cao Huaize 4 caohuaize@qq.com
Xiong Wei 1 xiongwei@dali.edu.cn
1 College of Basic Medical Sciences, Dali University , Dali, Yunan , China
2 Department of Radiology, The People’s Hospital of Lincang City , Lincang, Yunnan , China
3 Department of Imaging Function, Maternal and Child Health Hospital of Dali Bai Autonomous Prefecture , Dali, Yunnan , China
4 Department of Radiology, The First Affiliated Hospital of Dali University , Dali, Yunnan , China
5 College of Life Sciences, Shaanxi Normal University , Xi an, Shaanxi , China
Uversky Vladimir
Electronic publication date: 2025 Nov 13
Publication date: 2025
Volume: 13
Electronic Location ID: e20274
Received 2025 May 22; Accepted 2025 Sep 30
Copyright: © 2025 Zhao et al.
Copyright year: 2025
Copyright holder: Zhao et al.
License: This is an open access article distributed under the terms of the Creative Commons Attribution License, which permits unrestricted use, distribution, reproduction and adaptation in any medium and for any purpose provided that it is properly attributed. For attribution, the original author(s), title, publication source (PeerJ) and either DOI or URL of the article must be cited.
License URL: https://creativecommons.org/licenses/by/4.0/

Keywords: LOXL3 protein, Lysyl oxidase, Collagen, Malignant tumors

Funding: National Natural Science Foundation of China 82160516 and 32160167 Yunnan Provincial Fundamental Research Projects 202201AT070004, 202301AT070023 Key Program of Yunnan Provincial Fundamental Research Projects 202001BB0500080 Yunnan Provincial Ten Thousand Talent Projects 2019 Yunnan Provincial Local University Joint Projects 202101BA070001-282, 202101BA070001-128, 202401BA070001-093 Scientific Research Fund of Yunnan Provincial Department of Education 2022J0688, 2022J0716 and 2022Y808 Dali City Science and Technology Planning Projects 2021KBG032 Open Projects of Yunnan Provincial Key Laboratory of Entomological Biopharmaceutical R&D AG202203, AP2022006 Second Batch of Discipline Construction Projects of the First Affiliated Hospital of Dali University DFYXK2023019 Xing-Guo Liu Expert Workstation in Dali Bai Autonomous Prefecture 202402 This research was funded by the National Natural Science Foundation of China (Nos. 82160516 and 32160167), Yunnan Provincial Fundamental Research Projects (Nos. 202201AT070004, 202301AT070023), Key Program of Yunnan Provincial Fundamental Research Projects (No. 202001BB0500080), Yunnan Provincial Ten Thousand Talent Projects (No. 2019), Yunnan Provincial Local University Joint Projects (Nos. 202101BA070001-282, 202101BA070001-128, 202401BA070001-093), Scientific Research Fund of Yunnan Provincial Department of Education (Nos. 2022J0688, 2022J0716 and 2022Y808), the Dali City Science and Technology Planning Projects (No. 2021KBG032), Open Projects of Yunnan Provincial Key Laboratory of Entomological Biopharmaceutical R&D (Nos. AG202203, AP2022006), the Second Batch of Discipline Construction Projects of the First Affiliated Hospital of Dali University (DFYXK2023019), and the Xing-guo Liu Expert Workstation in Dali Bai Autonomous Prefecture (No. 202402). The funders had no role in study design, data collection and analysis, decision to publish, or preparation of the manuscript.

==============================
Lysyl oxidase-like 3 (LOXL3) is a key member of the lysyl oxidase (LOX) family and belongs to the copper-dependent amine oxidase family. Its traditional core function is to catalyze the cross-linking of collagen and elastin in the extracellular matrix (ECM), thereby maintaining the structural integrity and normal physiological functions of the ECM. In recent years, studies on cancer molecular mechanisms have confirmed that LOXL3 exhibits abnormal expression in a variety of cancers: in common malignant tumors such as melanoma, liver cancer, gastric cancer, colorectal cancer, and breast cancer, its expression level is significantly higher than that in the corresponding normal tissues. Meanwhile, numerous prognostic analyses have demonstrated that high LOXL3 expression is an independent risk factor for poor prognosis in cancer patients. Such patients usually have shorter progression-free survival (PFS) and overall survival (OS), suggesting that LOXL3 may serve as a potential biomarker for evaluating cancer prognosis. At the functional and mechanistic level, the role of LOXL3 is not limited to ECM remodeling. It can directly affect key biological behaviors of cancer cells, including proliferation, invasion, metastatic potential, and sensitivity to chemotherapeutic drugs, by regulating a variety of intracellular signaling pathways. This article reviews the specific roles and potential molecular mechanisms of LOXL3 in cancer, covering its associations with key cancer pathological processes such as epithelial-mesenchymal transition, maintenance of genomic stability, and regulation of the tumor microenvironment. It focuses on clarifying the specific molecular pathways through which LOXL3 promotes pro-tumor activities in different tumors, as well as the regulatory effects of these pro-tumor activities on patients’ relevant prognosis.

Introduction

Lysyl oxidase-like 3 (LOXL3) is a copper-dependent amine oxidase and a member of the lysyl oxidase (LOX) family, which includes LOX and lysyl oxidase-like 1-4 (LOXL1-4) (Santamaria et al., 2022). The core function of this protein family is to catalyze the cross-linking of collagen and elastin within the ECM, thereby maintaining tissue homeostatic balance and structural integrity. This process of ECM remodeling plays an indispensable role in various physiological and pathological processes, including embryonic development, tissue repair, and fibrosis (Di Mauro et al., 2024). LOXL3 is a multifunctional secretory protein, characterized most distinctly by the presence of four N-terminal scavenger receptor cysteine-rich (SRCR) domains and a highly conserved C-terminal catalytic domain (Santamaria et al., 2022). Furthermore, LOXL3 possesses unique glycosylation modification characteristics, including three potential O-linked glycosylation sites and five N-linked glycosylation sites, as well as two major transcript variants (LOXL3-sv1 and LOXL3-sv2). These structural features lay the foundation for LOXL3 to exert multiple roles in physiological and pathological processes. Beyond its basic amine oxidase activity, LOXL3 has additional functions. For example, it can catalyze the oxidation of fibronectin; this process activates the integrin signaling pathway, which in turn regulates key biological behaviors such as cell adhesion and migration (Maki & Kivirikko, 2001). Additionally, LOXL3 can act as a bispecific enzyme involved in the regulation of signal transducer and activator of transcription 3 (STAT3) acetylation and deacetylation to control inflammatory responses (Kraft-Sheleg et al., 2016). Moreover, studies suggest that LOX and LOXL3 are associated with “collagen trimerization” and “the assembly of collagen fibers and other multimeric structures,” while LOX, LOXL1, and LOXL2 are related to “elastic fiber formation” (Zhao et al., 2018). There are also studies showing that LOXL3 knockdown impairs the production of immature divalent cross-links in collagen fibers (Park et al., 2024).

Under normal physiological conditions, LOXL3 participates in processes such as embryonic development, bone formation, and neural differentiation, contributing to tissue microenvironment homeostasis by modulating ECM remodeling. However, recent studies have revealed that LOXL3 is frequently overexpressed in multiple types of malignant tumors, and its expression level is closely correlated with advanced tumor stage, lymph node metastasis, and poor patient prognosis (Laurentino et al., 2019). Unlike other members of the LOX family, which focus on “ECM remodeling”, LOXL3 can also participate in tumorigenesis and development by regulating epithelial-mesenchymal transition (EMT), interfering with cell signaling pathways, mediating chemotherapy resistance, and maintaining genomic stability, and shows certain specificity in different tumor types (Laurentino et al., 2019). An in-depth analysis of the mechanisms by which LOXL3 operates across different tumors will not only offer novel insights into tumor heterogeneity, but also provide a theoretical foundation for developing precision therapeutic strategies targeting LOXL3. This review systematically evaluates current knowledge regarding the structural and functional characteristics of LOXL3, with a specific focus on recent discoveries implicating its role in human carcinogenesis. By integrating currently available information, we aim to lay a theoretical groundwork for further investigation into LOXL3 as a potential prognostic biomarker, therapeutic target, and diagnostic marker in cancer.

The intended audience

The primary target audience of this review is researchers in the fields of tumor and the LOX family. This article primarily reviews the structure of the LOXL3 gene and protein, as well as the tissue expression of the LOXL3 protein, its subcellular localization, amine oxidase activity, and regulatory mechanisms. It specifically focuses on elucidating how LOXL3 contributes to pro-cancer activities across various tumors and its implications for related prognostic outcomes.

Survey Methodology

To comprehensively summarize the research progress in the field of LOXL3 and cancer, we conducted a systematic literature review using two major databases, PubMed and Web of Science, as the retrieval platforms. Core search terms “LOXL3” and “Cancer” were combined using boolean operators for the literature search. In the design of the search strategy, no restrictions were imposed on the type of publications, and all literatures in this research field that contain the above-mentioned key concepts were covered. Subsequently, through a rigorous literature screening process—first, irrelevant literatures were initially excluded based on titles and abstracts, and then the full texts of potentially relevant literatures were carefully evaluated—we finally identified and included 77 relevant articles that met the inclusion criteria and were published between April 2001 and July 2025 into the in-depth analysis scope of this review.

Structure of LOX family

The LOX family constitutes a class of copper-dependent amine oxidases. All LOX family members share a conserved C-terminal region that includes a copper-binding site essential for enzymatic activity, a lysine tyrosylquinone (LTQ) cofactor that forms the catalytic core, and a cytokine receptor-like (CRL) domain whose function has not yet been fully clarified (Liu, Wang & Liu, 2025). This highly conserved catalytic domain catalyzes the oxidative deamination of ε-amino groups in collagen and elastin molecules, generating corresponding aldehydes. These aldehydes subsequently undergo spontaneous aldol condensation reactions, forming covalent cross-links. This process enhances the mechanical stiffness of the ECM (Ye et al., 2020a). Based on the heterogeneity of their N-terminal sequences, the LOX family can be classified into two distinct subfamilies. The first subfamily, which includes LOX and LOXL1, undergoes proteolytic processing by peptidases, generating the active enzyme and an N-terminal propeptide. The second subfamily comprises three members: LOXL2, LOXL3, and LOXL4. A notable feature of these members is that their N-terminal regions are enriched with SRCR domains (Barker, Cox & Erler, 2012, Nishioka, Eustace & West, 2012, Mayorca-Guiliani & Erler, 2013). These domains are likely to mediate protein-protein interactions and influence processes such as cell adhesion, signal transduction, and immune function (Sarrias et al., 2004, Reichhardt et al., 2020). For example, the SRCR domains of LOXL2 interact with type IV collagen and fibronectin (Umana-Diaz et al., 2020), and SRCR domain 1 is essential for interaction with RNA-binding protein (Eraso et al., 2023). The SRCR domains of LOXL3 can deacetylate STAT3 at multiple acetyl-lysine sites (Ma et al., 2017). Additionally, LOXL2 binds to histone H3, and both its amine oxidase and SRCR domains can catalyze the deacetylation of H3K36ac (Lu et al., 2022). In addition, sequence analysis reveals that LOXL3 shares 56.54% amino acid homology with LOXL2, but only about 50% homology with other family members (LOX, LOXL1, and LOXL4) (Table 1). The conserved catalytic core supports fundamental biological functions required for life, while structural “innovations” in the N-terminal region enable these proteins to adapt to the complexity of multicellular organisms. Through the acquisition of new functions, such as transcriptional regulation and involvement in specific signaling pathways, they finely modulate development, homeostasis, and disease progression (Fig. 1).

Table 1 Characterization of different members of the human LOX protein family.

LOX Isoform	Chromosomal location	Full-length cDNA (kb)	Numbers of amino acids	Homology with LOXL3 (%)	Distribution	
LOX	5q23.1	5.07	417	52.48	Fat, gallbladder, bladder, placenta, endometrium	
LOXL1	15q24.1	2.34	574	53.96	Heart, prostate, placenta, fat, bladder	
LOXL2	8q21.3	3.72	774	56.54	Placenta, fat, appendix, endometrium, bladder	
LOXL3	2p13.1	3.68	753	100.00	Placenta, bone marrow, spleen, appendix, adrenal	
LOXL4	10q24.2	3.59	756	55.23	Thyroid, testes, ovaries, endometrium, prostate	
Note:

The data were obtained from the National Center for Biotechnology Information (NCBI). The top five tissues in terms of gene expression levels in the NCBI database are shown.

Figure 1 Structural characteristics of lysyl-oxidases.

Created in: Adobe Photoshop 2024.

The structure of the loxl3 gene and protein

The human LOXL3 gene (Gene ID: ENSG00000115318) is explicitly localized at the chromosomal region 2p13.1. The full-length gene comprises a total of 23,462 nucleotides (approximately 23.5 kb) and contains 14 exons. These exons are spliced together through an RNA splicing process to form mature mRNA, with different exons encoding distinct functional domains of the protein (such as the catalytic domain and the cytokine receptor-like domain). The mature cDNA transcribed from the LOXL3 gene has a length of approximately 3,689 base pairs (bp). This transcript contains a complete open reading frame (ORF), which can be translated into the LOXL3 protein; it also includes a 5′ untranslated region (5′UTR) and a 3′ untranslated region (3′UTR). These regions are involved in post-transcriptional processes such as the regulation of mRNA stability and translational efficiency. The promoter of LOXL3 is the core of its transcriptional regulation. Embedded in the 5′-flanking (upstream) sequence of exon 1, the promoter lacks typical TATA boxes and CAAT boxes, and its transcriptional initiation relies on the synergistic action of other regulatory elements. Transcription factors such as STAT3 and signal transducer and activator of transcription 6 (STAT6), which are involved in inflammation and gene activation within the tumor microenvironment (Ma et al., 2017); serum response factor (SRF), which regulates cytoskeleton-associated genes; the general transcription activator transcription factor Sp1 (Sp1), which participates in basal transcription; paired box 6 (PAX6), which controls development-related genes; nuclear factor κappa B (NF-κB), which mediates inflammatory and immune responses; and c-Rel, an NF-κB family member regulating immune cell activation—all can bind to regulatory motifs in the LOXL3 promoter. The presence of these diverse regulatory elements enables LOXL3 to respond to a variety of cellular signals and modulate its expression levels under different physiological or pathological conditions, thereby fulfilling context-specific functions (Fig. 2A). The signal peptide, located at the extreme N-terminus of the protein, directs its translocation into the endoplasmic reticulum after synthesis, where it is subsequently cleaved off and glycosylation modifications occur. Extracellularly, BMP-1 cleaves LOXL3 at its GDD site, generating four SRCR domains and an intact catalytic domain. This cleavage process is critical for LOXL3 to function as a secreted protein. Similar to other members of the LOX family proteins, the catalytic domain of LOXL3 is located at the C-terminus, which serves as the core for LOXL3 to exert its amine oxidase function. LOXL3 protein can undergo glycosylation modifications. Studies have indicated that it contains three potential O-linked glycosylation sites (Ser-26, Ser-28, Ser-30), as well as five N-linked glycosylation sites located downstream of the signal peptide cleavage site. This glycosylation site pattern has not been observed in other LOX-like proteins. Furthermore, these glycosylation sites in LOXL3 are likely necessary for its efficient secretion and function. In conditioned medium from fibrosarcoma cells (HT-1080), recombinant LOXL3 protein exhibits an apparent molecular weight of 97.0 kDa, which is higher than the predicted value—a discrepancy that may be attributed to glycosylation modifications (Maki & Kivirikko, 2001).

Figure 2 Schematic diagram of the structure of the LOXL3 gene and the protein variants it encodes.

Created in: Adobe Photoshop 2024.

During the search for homologs of the LOXL3 gene expression sequence, two transcript variants of LOXL3, named LOXL3-sv1 and LOXL3-sv2, were identified in the human EST (Expressed Sequence Tag) database (Lee & Kim, 2006). LOXL3-sv1 represents the most extensively truncated splice variant, characterized by the absence of exons 1 through 3 and exon 5. Notably, a genomic region upstream of exon 4 contains an 80 bp alternative promoter element that drives constitutive transcription of this isoform. Additionally, a 561 bp sequence is present in the 3′ flanking region of exon 14, suggesting the presence of distinct regulatory elements within these genomic regions. The promoter of LOXL3-sv1 is capable of binding multiple transcription factors, including p53, GATA3 binding factor, Ras1 responsive binding protein, neurogenesis associated factors 1 and 3, activator protein-1 (AP-1), p21 (RAC1) activated kinase 2/5/8 (PAK2/5/8), interleukin-1 (IL-1), and their corresponding binding partners. In contrast, the LOXL3-sv2 variant consists of 12 exons. This isoform is composed of 608 amino acids with a predicted molecular mass of 67.4 kDa. In terms of post-translational modifications, it is predicted to undergo both O-linked and N-linked glycosylation and also contains a putative BMP-1 cleavage site (Jeong & Kim, 2017) (Fig. 2B).

Tissue expression and subcellular localization of loxl3 protein

Tissue expression of LOXL3 protein

The tissue distribution of LOXL3 expression level was obtained in the Human Protein Atlas (HPA) database (https://www.proteinatlas.org/). LOXL3 transcriptional activity peaks in colonic tissue, cardiac muscle, adipose depots, and splenic parenchyma, with transcript abundance ranking highest among these organs compared to other human tissues (Fig. 3A). Notably, overexpression of LOXL3 transcripts has been consistently observed in multiple malignancies, including adrenocortical carcinoma, invasive breast carcinoma, diffuse large B-cell lymphoma, esophageal carcinoma, glioblastoma multiforme, head and neck squamous cell carcinoma, clear cell renal carcinoma, papillary renal cell carcinoma, acute myeloid leukemia, lower-grade gliomas, ovarian serous cystadenocarcinoma, cutaneous melanoma, gastric adenocarcinoma, and testicular germ cell tumors (Fig. 3B). Moreover, the cBioPortal database (https://www.cbioportal.org/) uncovers that the LOXL3 gene is commonly affected by mutations and deletions in neoplastic tissues. Intriguingly, the incidences of gene mutations, amplifications, and deep deletions associated with LOXL3 exhibit marked variability across diverse cancer subtypes, highlighting the gene’s distinct mutational landscapes in different tumor contexts. Notably, the mutation frequency of LOXL3 is highest in uterine corpus endometrial carcinoma, whereas the frequency of deep deletions is highest in diffuse large B-cell lymphoma (Fig. 3C). Furthermore, comprehensive analysis of TCGA tumor samples reveals that the R375C substitution is the most prevalent point mutation in LOXL3 (Fig. 3D).

Figure 3 (A–D) Tissue-specific expression analysis of LOXL3.

Subcellular localization of the LOXL3 protein

Studies by Kraft-Sheleg et al. (2016) using LOXL3 knockout mouse models demonstrated that LOXL3 mediates oxidative modification of fibronectin in the ECM, with this regulatory effect being particularly pronounced at myotendinous junctions. This finding underscores the role of LOXL3 in altering the structural and functional properties of fibronectin within this specific tissue interface, potentially influencing tendon integrity and homeostasis (Kraft-Sheleg et al., 2016). In HeLa cells (human cervical cancer cells), melanoma cells, and Madin-Darby canine kidney (MDCK) cells (an epithelial cell model), immunofluorescence staining and western blot analysis revealed that LOXL3 protein is primarily localized in the cytoplasm, with significant perinuclear enrichment. Furthermore, the expression level of LOXL3 in these cells was higher than in normal control cells (Peinado et al., 2005, Santamaria et al., 2018). Additionally, gastric cancer cells have been shown to express LOX protein in both the cytoplasm and the nucleus (Kasashima et al., 2018). LOXL3 can be transported into the nucleus due to its nuclear localization signal (residues 293-311), which consists of two basic amino acid clusters (Maki, Tikkanen & Kivirikko, 2001). Functionally, following stimulation with oncostatin M (OSM), LOXL3 translocates to the nucleus. There, it participates in regulating STAT3-mediated downstream signaling pathways, such as those controlling cell proliferation and inflammatory responses, through the removal of lysine acetylation at the K685 site of STAT3 (Ma et al., 2017). Beyond its nuclear localization, LOXL3 exhibits a unique subcellular distribution pattern not commonly observed among other lysyl oxidase family members: it is partially localized to mitochondria. This mitochondrial targeting is dependent on the translocase of the outer membrane (TOM) complex located on the mitochondrial outer membrane. Serving as the general entry gate for mitochondrial proteins, the TOM complex facilitates the import of LOXL3 into the mitochondrial outer membrane or intermembrane space, where it is hypothesized to participate in processes related to redox homeostasis or energy metabolism (Rapaport, 2002). Complementary yeast two-hybrid screening and co-immunoprecipitation (Co-IP) experiments confirmed that LOXL3, as a secreted protein, is exported extracellularly and may potentially interact with human telomerase reverse transcriptase (hTERT) (Zhou et al., 2013). Meanwhile, it has been observed that LOXL3-sv1 is localized in the cytoplasm, while the same study also reported that the subcellular localization of LOXL3-sv2 protein is unstable (Lee & Kim, 2006) (Fig. 4).

Figure 4 Subcellular localization of LOXL3 protein.

Image credit source: the Human Protein Atlas database, https://www.proteinatlas.org/ENSG00000115318-LOXL3/subcellular.

Loxl3 protein amine oxidase activity

The cleavage site of BMP-1 is located in the propeptide region preceding the SRCR domain of LOXL3 (Fig. 1) (Jourdan-Le Saux et al., 2001). Following cleavage, the propeptide is removed, and the remaining portion constitutes the mature LOXL3 enzyme. This processing event occurs within the ECM and results in the production of a catalytically active amine oxidase, with a predicted molecular weight of 35.0 kDa for the C-terminal protein. However, western blot analysis has demonstrated that the protein cleaved from colon tissue exhibits a molecular weight of 40.0 kDa, a discrepancy potentially attributable to post-translational modifications such as glycosylation. Studies have shown that the LOXL3 protein and its derivatives possess amine oxidase activity, acting upon elastin and various collagen subtypes, specifically types I, II, III, IV, VI, VIII, and X (Fig. 3). Collagen VIII demonstrates higher reactivity with LOXL3, whereas collagen types I, IV, and X are more active with LOXL3-sv1 (Huang et al., 2001). Additionally, recombinant LOXL3-sv2 protein showed a significant level of amine oxidase activity toward collagen type I in a BAPN-sensitive manner (Jeong & Kim, 2017). Research further indicates that the amine oxidase inhibitor β-aminopropionitrile (β-APN) can effectively and irreversibly inhibit the amine oxidase activity of LOXL3, thereby preventing the formation of protein cross-links in the cellular matrix (Fig. 5) (Lee & Kim, 2006, Jeong & Kim, 2017).

Figure 5 LOXL3 protein amine oxidase activity.

Created in: BioRender.

The mechanism of action of loxl3 protein in the development of human malignancies

Transcription factors snail family transcriptional repressor 1 (SNAIL) is capable of inducing the EMT process through repressing the transcription of the E-cadherin (CDH1) gene. Previous studies have demonstrated that LOXL3 colocalizes with the transcription factor SNAIL in the perinuclear region. The underlying mechanism by which LOXL3 exerts its function involves enhancing the stability of SNAIL protein: specifically, LOXL3 impedes the proteasomal degradation pathway of SNAIL and modulates its nuclear export process, thereby reducing the turnover rate of SNAIL and preventing its premature degradation. The physical interaction between LOXL3 and SNAIL is essential for suppressing E-cadherin expression. As a key epithelial cell marker, E-cadherin is encoded by the CDH1 gene, and its downregulation is widely recognized as a central hallmark of EMT initiation and progression. Further investigations have revealed that the synergistic interplay between LOXL3 and perinuclear membrane-associated SNAIL leads to the cooperative inhibition of CDH1 transcriptional activity. This collaborative repression ultimately facilitates the acquisition of a mesenchymal phenotype in tumor cells, accompanied by enhanced invasive and metastatic potentials. Collectively, this regulatory mechanism establishes a direct functional link between LOXL3 and the processes of tumor progression and metastasis (Peinado et al., 2005). Additionally, protein kinase D1 (PKD1) regulates the phosphorylation of the Ser-11 residue on SNAIL protein, thereby enhancing the expression of LOXL3 and its association with SNAIL. Phosphorylation at the Ser-11 site of SNAIL facilitates the recruitment of histone deacetylases 1 and 2 (HDAC1 and HDAC2) together with LOXL3. The phosphorylated variant of SNAIL, in complex with HDAC1 and HDAC2, exhibits stability in the nuclear environment and promotes the upregulation of tumor proliferation markers, including CYCLIN D1 and the Ajuba LIM protein (AJUBA) (Fig. 6) (Jeong et al., 2018).

Figure 6 The role that the LOXL3 protein plays in human cancer development and metastasis.

Created in: BioRender.

Research has demonstrated that LOXL3 plays a role in the initiation and progression of melanoma. In the malignant transformation of melanocytes, LOXL3 overexpression correlates with the mechanism underlying the activation of B-Raf proto-oncogene (BRAF) signaling. Moreover, increased LOXL3 gene expression in melanoma is associated with reduced methylation levels in its promoter region. Furthermore, suppression of LOXL3 expression enhances DNA damage signals within melanoma cells, leading not only to an increase in DNA double-strand breaks (DSB) and defects in G2/M phase progression, but also to reduced cellular apoptosis. Additionally, LOXL3 interacts with several proteins critical for maintaining genomic integrity and supporting proper cell division, including BRCA2 DNA repair associated (BRCA2), RAD51 recombinase (RAD51), structural maintenance of chromosomes 1A (SMC1A), MutS Homolog 2 (MSH2), structural maintenance of chromosomes 3 (SMC3), and nuclear mitotic apparatus protein 1 (NUMA1) (Zhou et al., 2013). Transforming growth factor-β (TGF-β) signaling upregulates the transcriptional level of LOXL3 in gastric cancer cells. Mechanistically, LOXL3 serves as a critical downstream mediator in the TGF-β signaling cascade. Functionally, the activation of LOXL3 plays a central driving role in the malignant progression of gastric cancer, not only enhancing the metastatic capacity of cancer cells but also promoting the formation of an invasive phenotype (Jones et al., 2018).

Tumor-promoting effect of loxl3 on cancer

The role of LOXL3 in glioma

Glioma of the brain is caused by a primary malignancy of the brain tumor, and notorious for treatment resistance (Gusyatiner & Hegi, 2018). Xia et al. (2022) identified that LOX, LOXL1, LOXL2 and LOXL3 were highly expressed in glioma cell lines (T98G and A172) at the mRNA and protein levels. Glioblastoma (GBM) is the malignant tumor with the highest expression of LOXL3 across various tumor types. The expression of LOXL3 was consistently elevated in GBM specimens compared to normal brain parenchyma across all molecular subtypes, as demonstrated in recent studies by Laurentino and colleagues. Clinical data analysis demonstrated a significant inverse association between LOXL3 expression levels and survival outcomes, with elevated LOXL3 levels correlating with poorer prognostic indicators in patient cohorts, where elevated LOXL3 levels predicted reduced OS rates. In U87MG cells, inhibition of LOXL3 gene expression reduced cell proliferation and invasion ability, induced cell death, and enhanced cell adhesion. In addition, LOXL3 silencing leads to downregulation of the MAPK/ERK signaling cascade responsible for regulating transcription, translation, cytoskeletal rearrangement, and focal adhesion degradation (Laurentino et al., 2021). In gliomas, the overexpression of LOX/LOXL is closely associated with tumor progression and shows a strong correlation with poor survival rates in patients. Based on this finding, Xu et al. (2025) successfully developed a prognostic model for predicting the OS of patients by integrating the expression status of LOX/LOXL and their co-expressed genes.

Meanwhile, studies have also revealed that copper metabolism plays an important role in the regulation of glioma proliferation, angiogenesis, and TME. Further research indicates that copper metabolism-related genes are abnormally expressed in tumor tissues compared to normal tissues, making them potential prognostic markers for gliomas (Cazzoli et al., 2023). It is on the basis of this research that Li et al. (2023) constructed a prognostic model for glioma patients, which incorporates copper metabolism-related genes including LOXL3, providing valuable references for the diagnosis and treatment of glioma patients.

Astrocytoma (or cerebral astrocytoma) is a frequent tumor within the central nervous system. The expression levels of LOXL1 to LOXL4 demonstrate a gradual upregulation across grade II to IV astrocytomas, with the highest levels observed in GBM; these levels are markedly elevated compared to non-neoplastic brain tissues. Additionally, both LOXL1 and LOXL3 in glioma are implicated in the Wnt/β-catenin signaling pathway, and this signaling pathway plays an important role in regulating cell migration and apoptosis (Laurentino et al., 2022).

Generally, in glioma, LOXL3 is highly expressed, and its elevated expression is associated with poor prognosis. Knockdown of LOXL3 leads to reduced cell proliferation and migration. As an enzyme involved in collagen metabolism, LOXL3 participates in the regulation of collagen remodeling via the MAPK/ERK signaling pathway. Highly expressed LOXL3 may alter the structure and composition of the ECM, activate certain cell surface receptors, and subsequently trigger the MAPK/ERK signaling pathway. Activation of MAPK/ERK signaling promotes the expression of downstream target genes, influencing biological behaviors such as proliferation and invasion of glioma cells. As a copper-dependent amine oxidase, LOXL3 facilitates cross-linking of ECM components, including collagen. In glioma, LOXL3-driven ECM remodeling may create a favorable microenvironment for the activation of the Wnt/β-catenin signaling pathway, thereby promoting tumor cell proliferation and invasion. Further investigation into the relationship between LOXL3 and the Wnt/β-catenin signaling pathway in glioma will help elucidate the underlying mechanisms of gliomagenesis and provide new targets and insights for developing more effective therapeutic strategies.

The role of LOXL3 in lung cancer (LC)

Lung cancer (LC) remains a major tumor disease burden worldwide. Current epidemiological surveillance data show that there are more than 2 million new cases each year, and the global mortality rate is close to 1.76 million. It is worth noting that the ratio between mortality and incidence is imbalanced, which highlights the clinical severity of lung cancer (Thai et al., 2021). The LOX family genes (LOX, LOXL1, LOXL2, and LOXL3) exhibit elevated expression in a range of malignancies, including lung adenocarcinoma (LUAD). High expression of LOXL3 is associated with poor prognosis in patients. LOXL3 may regulate the secretion of cytokines and chemokines by tumor cells, recruit immunosuppressive cells into the tumor microenvironment, and inhibit the body’s anti-tumor immune response, thereby affecting tumor prognosis (Zheng et al., 2024). Database analysis results and cell function experiments show that the expression levels of LOXL2 and LOXL3 in LC patients are relatively high, and this high expression state indicates a poor prognosis for patients. Knockdown of LOXL2/LOXL3 can inhibit EMT, cell proliferation, migration, and invasion processes, while also enhancing the apoptosis of lung cancer cells. In addition, LOXL3 interacts with CCAAT/enhancer-binding protein α (CEBPA) and B-cell lymphoma 2 (BCL-2). When the expression of LOXL3 is downregulated, the ubiquitination level of BCL-2 in lung cancer cells increases, while its expression level decreases. CEBPA can recruit Tip60, thereby enhancing the transcription of LOXL2 and LOXL3 and the histone acetylation process in lung cancer cells. Finally, CEBPA binds to Tip60 to promote the transcription of LOXL2 and LOXL3, thereby enhancing the stability of BCL-2 and ultimately promoting the development and metastasis of lung cancer cells (Fan et al., 2024).

In summary, LOXL2 and LOXL3 are highly expressed in lung cancer and influence tumor prognosis by modulating immune infiltration. Knockdown of LOXL2/LOXL3 inhibits EMT, proliferation, migration, and invasion, while promoting apoptosis in lung cancer cells. In lung cancer, CEBPA enhances the transcriptional expression of LOXL2/LOXL3 and stabilizes BCL-2, thereby promoting the initiation and progression of lung cancer.

The role of LOXL3 in hepatocellular carcinoma (HCC)

HCC is a prevalent malignant tumor and constitutes one of the primary contributors to cancer-related mortality (Wang & Deng, 2023). LOXL3 plays an indispensable catalytic role in orchestrating ECM dynamic restructuring, while critically governing the molecular configuration of both collagen fibrils and elastin networks. Wang et al. (2021) demonstrated that the expression level of the LOXL3 gene is significantly higher in malignant tissues compared to normal tissues. In HCC, LOXL3 expression was positively correlated with increased infiltration of various immune cell subtypes and elevated activity of immune checkpoint-related genes. Moreover, high LOXL3 expression was associated with poor prognosis in HCC patients, and in tumor tissues from metastatic HCC cases, its expression was directly linked to tumor size and clinical stage (Wang et al., 2021). Studies have shown that TGF-β1 is a key regulatory factor of fibrosis and can significantly affect the expression of LOX (Bae et al., 2018). The study found that treatment with TGF-β1 enhanced LOXL3 protein expression in HCC cell lines (SMMC-7721 and Huh-7), promoted cell invasion, and reduced apoptosis. Mechanistically, LOXL3 silencing was shown to inhibit HCC invasion and EMT through the Snail1/USP4-mediated feedback loop and the Wnt/β-catenin signaling pathway (Li et al., 2022).

In terms of drug resistance, further studies have revealed that LOXL3 promotes chemoresistance in liver cancer by stabilizing dihydroorotate dehydrogenase (DHODH) through activation of the intramitochondrial adenylate kinase 2 (AK2)-LOXL3-DHODH pathway, thereby inhibiting mitochondrial ferroptosis. Moreover, combination therapy using low-dose oxaliplatin and leflunomide (a DHODH inhibitor) effectively suppressed HCC progression and extended survival in both S704D-LOXL3 knock-in mice and transplanted tumor models with S704A-LOXL3 (Fig. 7) (Zhan et al., 2023). In the search for potential drugs targeting LOX and LOXL3 in liver cancer, 30 compounds were identified. β-aminopropionitrile demonstrated the strongest interaction. Quinones and copper were also highlighted as promising candidates due to their well-established functional roles within the LOX proteins. Quinones form part of the redox cofactor in the catalytic domain of LOXs. Furthermore, all LOX family members contain a conserved copper-binding site in their C-terminal region. Copper coordination by key histidine residues facilitates the formation of the quinone-containing cofactor, which is essential for oxidase activity. Therefore, both quinones and copper represent foundational targets for future research into their practical or therapeutic applications. Furthermore, several established anticancer agents, including cetuximab, bleomycin, cisplatin, and paclitaxel, were identified in the screening, highlighting their significant role in cancer treatment, particularly for liver cancer (Sun et al., 2022).

Figure 7 Diagrammatic representation of the AK2/LOXL3/DHODH axis-mediated resistance to ferroptosis caused by chemotherapeutic drugs in liver cancer.

Created in: BioRender.

All in all, TGF-β1 can activate the transcription of LOXL3 through its classical signaling pathway (TGF-β1/Smad2/3). Additionally, TGF-β1 promotes collagen synthesis, while LOXL3 enhances the structural stability and stiffness of the ECM by catalyzing collagen cross-linking, thereby increasing the migration and invasion capabilities of HCC cells. During chemotherapy, LOXL3 stabilizes DHODH by activating the mitochondrial AK2-LOXL3-DHODH pathway, inhibits mitochondrial ferroptosis, and promotes chemoresistance in HCC. Aminopropionitrile has a strong effect in the treatment of HCC and can be used as a potential therapeutic drug targeting LOXL3-related pathways, providing a new intervention direction for improving chemoresistance and inhibiting tumor progression in HCC.

The role of LOXL3 in gastric cancer

Globally, gastric cancer (GC) is the fifth most prevalent malignancy, and it accounts for the third highest number of cancer-associated mortalities worldwide. Pathophysiological investigations have identified helicobacter pylori infection as an established pathogenic determinant, while epidemiological stratification reveals age-dependent progression patterns and sex-specific incidence disparities. Dietary analyses further delineate modifiable risks through excessive sodium consumption (≥5 g/day) and phytonutrient-deficient dietary patterns characterized by suboptimal fruit/vegetable intake (<400 g daily) (Smyth et al., 2020). LOXL3 is primarily localized within the nucleus and is linked to invasive behavior and unfavorable prognosis in GC (Laurentino et al., 2019). Research indicates that the expression levels of LOX, LOXL1, and LOXL3 are significantly elevated in GC stages 2 to 4 compared to both normal tissues and earlier stages of the disease (Wang et al., 2021). In stomach cancer tissues and cells, signs of increased regulation of LOXL3 expression can be observed (Chu, Huang & Pan, 2025). Inhibition of LOXL3 can trigger iron death in gastric cancer cells, and the specific iron death inhibitor Ferrostatin-1 (Fer-1) effectively eliminates the inhibitory effect of LOXL3 on the proliferation, metastasis, and angiogenesis of gastric cancer cells. Knockout LOXL3 can be used as an important step in triggering iron death and inhibiting aggressive progression of stomach cancer (Chu, Huang & Pan, 2025).

In a cohort study involving 597 patients treated with primary treatment for gastric cancer, the OS with positive expression of LOXL1, LOXL3, or LOXL4 was significantly shorter than with negative LOXL4 expression. When diffuse GC cells are very aggressive, LOXL3 and LOXL4 mRNAs are significantly elevated. Furthermore, TGF-β has been shown to enhance the expression of LOXL3 and LOXL4 while concurrently downregulating LOXL1 expression. These findings suggest that LOXL3 could function as a modulatory node in the TGF-β signaling axis, a critical pathway driving pathological mechanisms including metastatic dissemination and invasive cell behavior in malignant progression (Kasashima et al., 2018). Therefore, LOXL3 and its LOX family have emerged as promising prognostic biomarkers with translational potential, while simultaneously presenting novel therapeutic targets for precision anti-neoplastic clinical management strategies.

In summary, LOXL3 expression is upregulated in gastric cancer cells and tissues, and its high expression is associated with poor prognosis. Knockdown of LOXL3 induces ferroptosis in gastric cancer cells, and the specific ferroptosis inhibitor Fer-1 effectively reverses the suppressive effects of LOXL3 knockdown on proliferation, metastasis, and angiogenesis. Additionally, the TGF-β signaling axis is a key pathway driving pathological mechanisms such as metastatic dissemination and invasive cellular behavior during malignant progression. LOXL3 acts as a downstream target of this pathway and enhances cell migration and invasion capabilities.

The role of LOXL3 in colorectal cancer

Colorectal cancer (CRC) ranks as the third most prevalent malignancy globally by incidence rate, constituting a major public health burden in oncological epidemiology; it concurrently accounts for the second foremost contributor to oncological mortality, maintaining its substantial epidemiological burden across global populations (Baidoun et al., 2021). Patients with high LOXL3 expression often present with increased lymph node metastasis and reduced survival rates. To evaluate treatment response in metastatic colorectal cancer (mCRC), a study proposed combining Zinc Finger E-Box Binding Homeobox 2 (ZEB2) and LOXL3 as circulating tumor cell (CTC)-based indicators of EMT (Insua et al., 2017). Clinically, desflurane is widely used in surgical anesthesia due to its high potency and low toxicity. However, studies have confirmed that under desflurane exposure, the miR-34a/LOXL3 axis promotes EMT and metastatic dissemination in CRC. Therefore, the clinical application of miR-34a mimics or inhibitors targeting LOXL3 may offer potential protective effects for CRC patients undergoing desflurane anesthesia (Ren et al., 2021). Barbazan et al. (2014) assessed the prognosis of patients with metastatic colorectal cancer (mCRC) using six prognostic markers, including LOXL3, and the clinical utility of prognostic tools, the results showed that the survival time of patients with high CTC markers was significantly shortened.

In summary, in colorectal cancer, high expression of LOXL3 is an important indicator of poor prognosis. Among relevant factors, ZEB2 acts as a core transcription factor in EMT, while LOXL3 mainly strengthens the EMT phenotype by remodeling the microenvironment and activating signaling pathways. Together, they enhance the metastatic potential of CTCs. Therefore, the combined monitoring of these two factors can dynamically reflect the EMT status of colorectal cancer patients, providing an effective basis for early warning of metastasis risk. In addition, in the clinical scenario of using desflurane for anesthesia, studies have found that miR-34a mimics or inhibitors targeting LOXL3 may play a potential protective role in colorectal cancer patients undergoing desflurane anesthesia, offering a new direction for optimizing the treatment regimens of such patients.

The role of LOXL3 in pancreatic cancer

Pancreatic cancer (PC) ranks seventh among principal contributors to worldwide oncological mortality, this epidemiological positioning underscores its substantial disease burden while imposing significant clinical challenges across diverse healthcare systems. Pancreatic ductal adenocarcinoma (PDAC) accounts for approximately 90% of histologically confirmed pancreatic malignancies, representing the predominant pathological subtype with distinct molecular and clinical profiles in gastrointestinal oncology (Stoffel, Brand & Goggins, 2023). Compared to normal tissues, PDAC tissues have increased levels of LOXL3 protein expression. It has been proposed that LOXL3 regulates immune cell recruitment and influences the expression of PDAC’s immunological traits (Jiang et al., 2022). Laurentino et al found, by interacting with SNAIL, LOXL3 causes pancreatic ductal adenocarcinoma cells to undergo EMT, which encourages cell division and metastasis (Laurentino et al., 2019). Contemporary oncological pathology has established that desmoplastic stroma remodeling—characterized by excessive ECM deposition—constitutes a pathognomonic histopathological hallmark of pancreatic ductal adenocarcinoma, while serving as a critical determinant in creating immunosuppressive microenvironments that substantially compromise the therapeutic responsiveness to immune-modulating treatment strategies. According to Zhao et al. (2024) fibrosis may be impacted by low concentrations of arsenic trioxide (ATO, 1.0 μm), which can also prevent pancreatic stellate cells (PSCs) activation. By altering ECM structure and promoting CD8+ T cell infiltration, low-dose Atos regulates LOXL3, enhancing the impact of immunotherapy and offering a new approach to PC treatment (Zhao et al., 2024).

Fibrosis formation is a key factor in the development of pancreatic cancer. PDAC can interact with the EMT-related transcription factor SNAIL, regulating EMT to promote pancreatic cancer metastasis. Excessive matrix deposition leads to fibrosis, resulting in a strongly immunosuppressive immune microenvironment, which contributes to poor response to immunotherapy. However, ATO may reverse the immunosuppressive microenvironment by improving the structure of the ECM, reducing the infiltration of immunosuppressive cells, and enhancing T-cell function. It can also regulate LOXL3, thereby boosting the efficacy of immunotherapy.

The role of LOXL3 in breast cancer

Breast cancer (BC) is a malignant tumor originating from the epithelium of mammary ducts or lobules. As the most commonly diagnosed cancer among women worldwide, its pathogenesis is associated with multiple factors such as genetics, hormones and the environment (Hong & Xu, 2022). Studies have shown that the expression levels of LOXL1, LOXL2, and LOXL3 are elevated in breast cancer tissues compared to normal tissues (Ramos et al., 2022). Among these, LOXL3 is closely associated with cancer cell proliferation, invasion, and metastasis. It interacts with SNAIL, thereby inducing invasive behavior in cancer cells (He et al., 2024). In MDA-MB-231 cells, knockout of LOXL1-LOXL4 resulted in a gradual decrease in cell migration activity (Liburkin-Dan et al., 2022). LOXL3 is commonly found in primary breast tumor sites and pleural effusions, with its expression level in effusions being lower than that in primary tumors (Sebban, Davidson & Reich, 2009). Furthermore, the expression of LOX mRNA and its related family members appears to be associated only with highly invasive or metastatic phenotypes of breast cancer cells, with no such correlation observed in non-invasive or non-metastatic cells (Kirschmann et al., 2002), such as in triple-negative breast cancer (Li et al., 2024). Immunohistochemical analysis by Jeong et al. (2018) revealed that LOXL3 protein expression is correlated with intratumoral and peritumoral inflammatory responses, as well as with estrogen receptor (ER), progesterone receptor (PR), and various molecular subtypes, but showed no association with patient prognosis. Notably, the collective migration of ductal breast adenocarcinoma cells through collagen-rich stromal barriers is essentially dependent on LOXL3-mediated enzymatic activity. This tumor-autonomous molecular driver promotes collagen fiber alignment via lysyl oxidase catalysis, resulting in a biomechanically reinforced microenvironment that enhances cooperative tumor cell invasion. Koorman et al. (2022) investigated the localization of LOXL3 using immunofluorescence and found that it is primarily expressed in myoepithelial cells surrounding normal human breast tissue and invasive ductal carcinoma of no special type (IDC-NST), with minimal expression in the cancer cells themselves. This finding highlights the role of LOXL3 in facilitating collective invasion of ductal breast cancer cells into collagen matrices by participating in collagen remodeling. Subsequent analysis of a clinical tissue microarray (TMA) containing 368 invasive breast cancer samples revealed cytoplasmic expression of LOXL3 in approximately 26.0% (76/292) of breast cancer patients (Koorman et al., 2022). Further studies indicated that LOXL3 is specifically expressed in breast myoepithelial cells and is focally upregulated in CK14-positive cells in contact with collagen. Although LOXL3 expression is detectable in high-grade invasive breast cancer, its levels show no significant correlation with tumor size, proliferative activity, or lymph node involvement (Li et al., 2024).

LOXL3 is primarily expressed in the myoepithelial cells of normal human breast tissue and in invasive ductal carcinoma of no special type. Its high expression is closely associated with cancer cell proliferation, invasion, and metastasis. LOXL3 interacts with transcription factors such as SNAIL to regulate EMT, thereby promoting malignant phenotypes and enhancing invasive and metastatic capabilities through ECM remodeling. The expression of LOXL3 is also correlated with intratumoral and peritumoral inflammatory responses, as well as ER, PR, and various molecular subtypes. Further studies are needed to elucidate the underlying mechanisms.

The role of LOXL3 in ovarian cancer

Ovarian cancer (OC) is clinically defined as a highly aggressive neoplasm predominantly detected at advanced clinical stages (Konstantinopoulos & Matulonis, 2023); although initial treatment with frontline platinum-based chemotherapeutic agents (e.g., carboplatin/paclitaxel) demonstrates marked chemosensitivity, this therapeutic susceptibility frequently diminishes with disease progression due to acquired platinum resistance mechanisms. Complete trypsin peptide or complete trypsinphosphopeptide testing revealed that ovarian cancer-specific proteins include the LOXL3 protein. The expression level of LOXL3 protein in the plasma of OC patients has been proven to be significantly higher (Dufresne et al., 2018). Overexpression of LOX, LOXL1, LOXL2, and LOXL3 mRNA indicated poor OS and PFS in OC patients, particularly in serous and grade II + III OC patients, and higher expression of LOXL3 may predict a worse clinical prognosis for patients with OC (Ye et al., 2020b).

In serous ovarian cancer, elevated expression levels of LOXL3 indicate a poor prognosis. OC develops resistance to chemotherapeutic drugs such as platinum-based agents and paclitaxel. We hypothesize that the possible reason is that ovarian cancer cells with high LOXL3 expression can enhance the interaction between the ECM and cells (e.g., through the activation of integrin signaling), thereby improving the resistance to chemotherapy-induced apoptosis and thus showing an enhanced survival advantage.

The role of LOXL3 in melanoma

Melanoma is a malignant tumor originating from melanocytes, primarily associated with ultraviolet radiation exposure. Frequent sun exposure increases the risk of developing melanoma (Ahmed, Qadir & Ghafoor, 2020). Both LOXL2 and LOXL3 are commonly expressed in melanoma cell lines (Kielosto et al., 2018). Studies have shown that alterations in melanoma phenotype in mice are partly mediated through the regulation of epithelial-mesenchymal transition transcription factors (EMT-TFs), such as SNAIL1 and PRRX1. The LOXL3-SNAIL1-PRRX1 axis may promote the acquisition of malignant phenotypes in melanoma, particularly in the presence of oncogenic BRAF mutations, thereby enhancing tumor cell invasion and migration (Santamaria et al., 2018). Consequently, LOXL3 has been proposed as a potential therapeutic target for melanoma treatment (Hajdu et al., 2018).

Immunohistochemical analysis has revealed that LOXL3 serves as a novel prognostic marker in primary melanoma. Studies indicate that elevated LOXL3 expression is positively correlated with increased tumor thickness and mitotic activity. In primary melanoma, high LOXL3 expression not only associates with tumor progression and invasiveness but also serves as an independent predictor of poor clinical outcomes. After adjusting for potential confounders using multivariate Cox regression analysis, Zhang et al. (2021) further confirmed that LOXL3 gene expression is an independent predictor of survival and remains a significant prognostic factor in a cohort of primary melanoma patients. As a prognostic marker reflecting the degree of tumor differentiation, LOXL3 is a major target of YTH domain family protein 3 (YTHDF3). YTHDF3 facilitates the binding of eukaryotic translation initiation factor 3 subunit A (eIF3A) to LOXL3 transcripts, thereby enhancing its protein expression without altering mRNA levels. Inhibition of LOXL3-mediated enzymatic activity significantly impairs the invasive and metastatic capacity of melanoma cells. Conversely, overexpression of LOXL3 reduces the suppressive effect of YTHDF3 knockdown on melanoma metastasis. The YTHDF3-LOXL3 axis represents a promising therapeutic target for inhibiting melanoma metastasis (Shi et al., 2022).

In vivo studies by Santamaria et al. (2018) demonstrated that LOXL3 cooperates with oncogenic BRAF to stimulate tumor growth and accelerate melanocyte transformation, suggesting its oncogenic role in melanoma development. Vazquez-Naharro et al. (2022) established a conditional LOXL3 knockout melanoma mouse model with a BrafV600E activating mutation and Pten deletion background. Their study revealed that knock down of LOXL3 in melanocytes significantly prolonged melanoma latency, suppressed tumor growth, and markedly reduced lymph node metastasis. Consistent with observations in human melanoma, LOXL3 also contributes to maintaining genomic stability in melanoma-derived cells in the mouse model (Vazquez-Naharro et al., 2022). Research by Laurentino et al. (2019) indicated that the upregulation of LOXL3 in melanoma is associated with demethylation of its promoter region. LOXL3 can interact with multiple proteins that play essential roles in maintaining DNA stability and promoting mitosis, thereby driving melanoma progression and sustained proliferation. Conversely, inhibition of LOXL3 activity triggers a DNA damage response (DDR) in melanoma cells, leading to accumulation of DSB, mitotic defects, G2/M cell cycle arrest, and ultimately programmed cell death (Laurentino et al., 2019). Thus, deficiency of LOXL3 results in accumulated DNA damage, impairing tumor cell proliferation. These findings suggest that targeted inhibition of LOXL3 in vivo may prolong OS in melanoma mouse models. Another study showed that LOXL3, SNAI1, and nestin (NES) are key mediators in melanoma pathogenesis, collaboratively regulating tumor progression, metastasis, and differentiation. It was found that peri-nuclear co-localization of LOXL3 and SNAI1 was consistent across all tumor subtypes, while NES and SNAI1 were co-expressed in melanoma cell populations. These results indicate a potentially dependent or synergistic mechanism among these markers during EMT in mesenchymal tumors, offering opportunities for developing novel therapeutic strategies. Further exploration and targeting of these mechanisms may lead to more precise treatment approaches (Situm Ceprnja et al., 2024) (Table 2). Regarding clinical applications, Xu et al. (2025) employed machine learning to identify gemcitabine as a potential therapeutic agent targeting LOX/LOXL overexpression in cutaneous melanoma; however, this finding has yet to be validated due to a lack of RNA-seq data.

Table 2 The carcinogenic function of LOXL3 across multiple human cancer.

Cancer types	LOXL3 expression	Effect observed	Target	References	
Glioma	Up regulated	Silencing LOXL3 downregulated MAPK/ERK	MAPK/ERK	Laurentino et al. (2021)	
β-catenin increases with tumor malignancy grade and is associated with the expression of LOXL3. Higher LOXL3 expression levels linked to a lower overall survival rate.	β-catenin	Laurentino et al. (2022)	
Lung cancer	Up regulated	LOXL3 knockdown results in decreased BCL-2 expression and increased BCL-2 ubiquitination suppressing EMT, proliferation, migration, and invasion.	BCL-2	Fan et al. (2024)	
Hepatocellular carcinoma	Up regulated	LOXL3 expression regulated by TGF-β1 to
promote tumorigenicity.	LOXL3-Wnt/β-catenin/Snail1	Li et al. (2022), Bae et al. (2018)	
LOXL3 promoted HCC progression via promotion of Snail1/USP4-mediated EMT	LOXL3-Snail1/USP4	Li et al. (2022)	
By triggering the activation of the AK2-LOXL3-DHODH axis in mitochondria, LOXL3 contributes to chemotherapy-induced ferroptosis.	AK2-LOXL3-DHODH	Zhan et al. (2023)	
Gastric cancer	Up regulated	Silencing LOXL3 might act as an inducer of ferroptosis, thereby impeding the progression of GC.	NA	Chu, Huang & Pan (2025)	
LOXL3 promotes the proliferation through the TGF-β signaling pathway	TGF-β1	Kasashima et al. (2018)	
Colorectal cancer	Up regulated	EMT progression and metastatic spread in CRC were potentiated by desflurane through miR-34a/LOXL3 axis	miR-34a/LOXL3	Ren et al. (2021)	
Pancreatic cancer	Up regulated	By regulating LOXL3, low-dose ATO is able to reshape the ECM, which in turn enhances the infiltration of CD8+ T cells	NA	Zhao et al. (2024)	
Breast cancer	Up regulated	LOXL3 correlates with the proliferation, invasion, and metastasis of cancer cells and interacts with SNAIL, contributing to invasive behaviors.	NA	He et al. (2024)	
Ovarian cancer	Up regulated	Among stage I + II OC patients, elevated levels of LOXL3 mRNA expression predicted inferior OS and PFS.	NA	Ye et al. (2020b)	
Melanoma	Up regulated	The LOXL3-SNAIL1-PRRX1 axis is involved in the phenotypic transition of melanoma and promotes tumor development.	LOXL3-SNAIL1-PRRX1	Vazquez-Naharro et al. (2022)	
When LOXL3 is overexpressed, it undoes the suppression of melanoma metastasis induced by the reduced expression of YTHDF3.	YTHDF3-LOXL3	Shi et al. (2022)	
		LOXL3 protein has a cancer-promoting function by preserving genomic stability and mitosis completion.	NA	Santamaria et al. (2018)	
LOXL3, SNAI1, and NES play crucial roles in melanoma development. These factors interact during the EMT process, jointly regulating tumor growth, metastasis, and cell differentiation.	NA	Situm Ceprnja et al. (2024)	
Note:

NA, Not available.

In summary, both LOXL2 and LOXL3 are frequently overexpressed in melanoma. Knockdown of LOXL3 reduces tumor cell proliferation and decreases lymph node metastasis. Mechanistically, LOXL3 is involved in the DDR pathway, thereby influencing cellular processes such as proliferation and migration. Furthermore, the LOXL3-SNAIL1-PRRX1 axis may promote the acquisition of malignant phenotypes in melanoma, affecting tumor cell proliferation and migration capabilities. The YTHDF3-LOXL3 axis also represents a potential therapeutic target for melanoma treatment. Finally, the interplay between multiple key mediators, including LOXL3, during melanomagenesis presents novel opportunities for the development of targeted therapeutic strategies.

Conclusions and prospects

LOXL3 demonstrates significant upregulation in various malignant tumors and plays a critical role in orchestrating core oncogenic processes during tumorigenesis and progression. Notably, its functional dominance manifests in driving invasive and metastatic cascades through multiple synergistic mechanisms. The enzyme enhances neoplastic cell migration and invasion primarily via potent EMT induction, while concurrently activating pivotal oncogenic signaling pathways including the Wnt/β-catenin axis and PI3K/AKT cascade, thereby accelerating malignant transformation. Mechanistic investigations reveal LOXL3’s capacity to remodel the tumor immune microenvironment through dual modulation of immune cell recruitment and cytokine network dynamics. Furthermore, evidence indicates its involvement in maintaining genomic stability and regulating the process of mitotic chromosome segregation. Systematic molecular profiling across cancer types has confirmed the predominantly pro-tumorigenic functions of LOXL3, and multi-cohort clinical validations have established its strong prognostic value. These collective findings establish LOXL3 as a promising prognostic biomarker and a critical therapeutic target, with significant potential to advance precision oncology through targeted inhibition.

Recent studies on the LOXL3 protein have revealed that, beyond its canonical function as a copper-dependent amine oxidase, it plays diverse biological roles. LOXL3 is involved not only in embryonic development but also in the pathogenesis of various diseases such as collagenopathies and cancers, collectively underscoring its significance as a potential therapeutic target (Laurentino et al., 2019). Because their structures are similar, there is now a nonselective inhibitor of the LOX family called BAPN. However, due to limited knowledge regarding individual LOX family members and their crystal structures, the development of specific LOX inhibitors has become an urgent and emerging challenge. The development of particular LOXL3 inhibitors for therapeutic applications was made possible by the newly released dual-action inhibitors of LOXL2/LOXL3 (Hajdu et al., 2018, Schilter et al., 2019). To achieve a comprehensive understanding of the function of LOXL3 in the oncogenesis and progression of malignant tumors, it is essential to address the current research limitations: existing studies have primarily focused on the downstream targets of the LOXL3 gene in tumor cell lines, while its upstream regulatory mechanisms and intrinsic degradation pathways remain poorly understood, warranting further investigation. It is also important to thoroughly examine the subcellular location of the LOXL3-sv1 and LOXL3-sv2 proteins in malignant tumors as well as the mechanism by which they contribute to tumor development. In addition, for instance, existing studies have confirmed that LOXL3 can be located in the nucleus, cytoplasm and mitochondria, but its nuclear localization signal (NLS) and mitochondrial targeting sequence (MTS) have not yet been identified. Moreover, it remains unclear whether its transport depends on companion proteins (such as the importin family). The solution to this problem may promote the development of intervention strategies targeting “LOXL3 subcellular localization”. Understanding the function of the LOXL3 gene in tumor formation and progression requires both substantial in vivo research and the use of engineered rodent models, such as knockout mice. In targeted research on LOXL3, on one hand, it is necessary to construct monoclonal antibodies that specifically bind to LOXL3, providing tools for subsequent detection, targeted delivery, and other applications. On the other hand, high-throughput screening technology can be used to identify low-molecular-weight inhibitors from a large number of compounds, which can exert inhibitory effects on certain human cancers with elevated LOXL3 gene expression. At the same time, the development of inhibitors that act directly on the LOXL3 protein is also crucial. These inhibitors can block the function of LOXL3 at the protein functional level, complementing the regulation at the gene level, and collectively providing support for the targeted therapy of related cancers.

Additional Information and Declarations

Competing Interests

The authors declare that they have no competing interests.

Author Contributions

Dan Zhao conceived and designed the experiments, prepared figures and/or tables, authored or reviewed drafts of the article, and approved the final draft.

Pu Su conceived and designed the experiments, prepared figures and/or tables, and approved the final draft.

Xuan Peng performed the experiments, authored or reviewed drafts of the article, and approved the final draft.

Xue Cheng performed the experiments, authored or reviewed drafts of the article, and approved the final draft.

Bin Li performed the experiments, authored or reviewed drafts of the article, and approved the final draft.

Xi-min Tang performed the experiments, prepared figures and/or tables, and approved the final draft.

Shaoyang Huang analyzed the data, prepared figures and/or tables, authored or reviewed drafts of the article, and approved the final draft.

Zhengliang Li analyzed the data, prepared figures and/or tables, and approved the final draft.

Huaize Cao conceived and designed the experiments, authored or reviewed drafts of the article, and approved the final draft.

Wei Xiong conceived and designed the experiments, authored or reviewed drafts of the article, and approved the final draft.

Data Availability

The following information was supplied regarding data availability:

This is a Literature Review.

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
