# Peer review of "The function and mechanisms of action of lysyl oxidase-like 3 (LOXL3) in cancers"

_PeerJ, doi:10.7717/peerj.20274_

## Round 0.1 · original submission · Major Revisions

· Academic Editor

Major Revisions

Please address the concerns of all reviewers and revise the manuscript accordingly.

**Language Note:** The review process has identified that the English language must be improved. PeerJ can provide language editing services - please contact us at [email protected] for pricing (be sure to provide your manuscript number and title). Alternatively, you should make your own arrangements to improve the language quality and provide details in your response letter. – PeerJ Staff

Reviewer 1 ·

Basic reporting

The review deals with the lysyl oxidase LOXL3, which takes part in enzymatic crosslinking of collagens, but also, through intricate mechanisms, has been shown to be transported into the nucleus and mitochondria, and in these locations to have functions independent from its ECM cross-linking extracellular function.

1. The MS is poorly organized and does not give a proper background about the gene, gene family, and phenotype of knockout mice (severe phenotype, basically raising major concerns about using it as a therapeutic target).

2. It is full of errors.

3. Writing is poor, and MS contains a lot of general statements, thrown in at places where they do not belong, like “breast cancer is a complex illness influenced by both environmental and genetic factors.”

4. It lacks logic and should not be published in its current form.

A few examples:
Line 28 Abstract: Variable linked---- should probably read “alternatively spliced”

Line 44- “family members are structurally identical” is not correct

Line 59 “section” should probably be “domain”

Line 63: What is 63% commonality with regard to proteins?

Line 146 “variation” should be “variant”

Line 175 LOXL3 radiated in the media---- cells mixed with cytoplasm and tissue matrix----neither expression makes sense.

Line 197, the protein SCR- “has a BMP-1 chain”- should read “BMP-1 cleavage site”.

Line 205 collagen exhibits amine oxidase activity for LOXL-2- the other way around, collagen is the substrate, not the enzyme.

Line 212v The transcription factor inhibited….. name of TF is lacking

Line 245 and onwards “tumor on the brain”… should read “brain tumor”

Line 390 pancreatic star cells – no such cell type exists.

Line 429: the frequency of the LOXL3 protein in blood plasma- might mean concentration

Line 477 – the gene name NES, encoding nestin, is wrongly interpreted as “neuroendocrine syndrome” (non-existing syndrome )
Line 505 fibrosis and collagen disease- fibrosis is a collagen disease

Line 519, Last sentence incomprehensible:
“Building monoclonal antibodies against LoxL3 and using high-throughput screening to find low molecular weight for the identification of certain human cancer inhibitors with elevated LOXL3 gene expression, LOXL3 protein inhibitors are also very crucial. “

Experimental design

-

Validity of the findings

-

Additional comments

More errors exist.

Reviewer 2 ·

Basic reporting

1. The abstract part of the article does not clearly explain the important role of LOXL3 in the occurrence and development of tumors and the theme of the article, nor does it state the key issues that this review wants to focus on. Each sentence feels like a separate part, and the conclusions come suddenly, without logic or regularity. It is suggested to sort out the key content of the full text and rewrite the abstract part of the article.

2. It is not in accordance with the academic writing standards to put the main results (Figure1 and Table1) directly in the introduction. I understand that the author wants to elicit the necessity and importance of in-depth research on LOXL3 through its special structure, but in fact, the desired effect is not achieved. As a review of a protein target, the introduction should clearly introduce the research status of this target in the field of cancer, cite key literature, outline the existing research progress, and then point out the gaps and deficiencies in the research field, and clarify the motivation and necessity of this review. And the difference should be specified and always review (such as https://doi.org/10.3390/ijms20143587), the last should have an objective conclusion about the targets you judge. Therefore, I think the author should rewrite the introduction part, and if the author thinks the content of Figure1 and Figure2 is necessary, the author can put it in the body part.

3. The chart is not standard. For example, ADAMTS cleavage site is not marked in Figure1, and the description of subcellular partition is lacking in Figure4. At the same time, the font and definition of the image in the whole article are not uniform (Such as, Figure3D mutation site labels are blurred), so the font size should be unified according to the PeerJ image guide. Tools such as Adobe illustrator can be used. Authors should further check the match between legend and article (for example, the position of Figure2B in the original article).

Experimental design

1. The methods section outlines the search strategy used by the authors, but fails to clearly describe the screening and selection process. The inclusion criteria, exclusion criteria, and filtering conditions must be explicitly stated to ensure transparency and reproducibility. The manuscript reports that 62 articles published over 16 years were included. However, the distribution of publication dates shows that most of the literature is older than five years. This raises concerns about the currency of the review. A preliminary search reveals several recent studies from the past two years that explore the role of LOXL3 in gastric cancer progression, pan-tumor mechanisms, and drug resistance (such as: DOI: 10.1186/s12964-025-02176-1, DOI: 10.1007/s12033-024-01229-z, DOI: 10.1615/CritRevEukaryotGeneExpr.2023049049); The authors must update the reference list to include recent publications to strengthen the scientific relevance and completeness of the review.

2. The logic of the article is weak. For example, although the roles of LOXL3 in glioma and melanoma are discussed respectively in parts 6.1 and 6.9, the common mechanisms (such as DNA repair) are duplicated, and there is no cross-reference.

3. At the same time, the elaboration depth of the mechanism of each cancer is insufficient, and it is not organized and summarized, which is more like a simple literature list. The mechanism of LOXL3 in the occurrence and development of tumors should be strengthened, and the authors need to think about and discuss it.

Validity of the findings

1. The authors documented the differential expression of LOXL3 across various tumors and organs utilizing data from the TCGA database, and they elaborated on the role of LOXL3 in distinct tumor types based on a literature review. However, the overall relationship between LOXL3 and clinical indicators, such as patient prognosis and response to drug treatment, remains inadequately elucidated.

2. The author appears to excessively rely on mechanistic inference. The interpretation of existing literature should prioritize the direct evidence presented within the studies, rather than solely depending on the inferred conclusions of the original works as a basis for citation.

Additional comments

1. Check the grammatical errors in the text, especially the subject-verb consistency;

2. Check the notation of the unit symbols in the text. For example, μm in line 389 should be μM.

3. Check the text for redundant expressions, such as 'due to the fact that'.

Reviewer 3 ·

Basic reporting

The manuscript "The function and mechanisms of action of lysyl oxidase-like 3 (LOXL3) in cancer" is a review that tries to address interesting trends in recent research on LOXL3, which is probably the least studied isoform of human lysyl oxidase. Potentially, authors' viewpoints may be useful for people working in various fields of basic molecular oncology and anti-cancer drug development. However, a number of excellent reviews on lysyl oxidases have already been published recently, which provided an in-depth analysis of published data on all five isoforms, their properties, and functions in health and disease. Thus, the scope of this article is not completely clear. Ideally, it should expose such properties of LOXL3 that make it unique among other isoforms. Unfortunately, the authors failed to do this; rather, they list various features of LOXL3 without a detailed comparison with other lysyl oxidases. Worst, the manuscript is poorly structured, and English looks like the text is a result of machine translation without human editing, but paradoxically with human-introduced typos like "secerted", "singaling", etc. All this gives an impression of a very raw draft that requires a major structural overhaul and heavy textual editing. One example illustrates the inaccuracy of wording: the enzyme is intermittently called either lysyl oxidase or lysinyl oxidase.

Figures also need to be improved. Some of them are illogical, for example, look at Fig. 4. The protein should be secreted, then what is it doing in intracellular compartments like mitochondria and the nucleus?

Many figures were copied and pasted from web portals like proteinatlas.org. It is not clear why prospective readers cannot simply look at these pics there; what is the reason to publish them in a scientific article?

Experimental design

The authors listed keyword searches; this may be adequate for a review paper. The cited literature appears to be insufficient to cover all important aspects.

Validity of the findings

Novelty of the conclusions in this review is doubtful. Perhaps it is necessary to stress that, previously, huge research efforts have been focused on LOXL2-specific drugs, whereas other isoforms were partially forgotten, including LOXL3. Indeed, recent discoveries indicate that inhibiting all five isoforms may be the key to success in oncology. However, in light of this, publication of a LOXL3-limited review is not justified.

---

## Round 0.2 · accepted · Accept

· Academic Editor

Accept

All issues pointed by the reviewers were addressed and the revised manuscript is acceptable now.

Reviewer 2 ·

Basic reporting

no comment

Experimental design

no comment

Validity of the findings

no comment